# Exogenous capture accounts for fundamental differences between pro- and antisaccade performance

Allison T Goldstein, Terrence R Stanford, Emilio Salinas*

Department of Neurobiology and Anatomy, Wake Forest School of Medicine, Winston-Salem, United States

**Abstract** To generate the next eye movement, oculomotor circuits take into consideration the physical salience of objects in view and current behavioral goals, exogenous and endogenous influences, respectively. However, the interactions between exogenous and endogenous mechanisms and their dynamic contributions to target selection have been difficult to resolve because they evolve extremely rapidly. In a recent study (Salinas et al., 2019), we achieved the necessary temporal precision using an urgent variant of the antisaccade task wherein motor plans are initiated early and choice accuracy depends sharply on when exactly the visual cue information becomes available. Empirical and modeling results indicated that the exogenous signal arrives ~80 ms after cue onset and rapidly accelerates the (incorrect) plan toward the cue, whereas the informed endogenous signal arrives ~25 ms later to favor the (correct) plan away from the cue. Here, we scrutinize a key mechanistic hypothesis about this dynamic, that the exogenous and endogenous signals act at different times and independently of each other. We test quantitative model predictions by comparing the performance of human participants instructed to look toward a visual cue or away from it under high urgency. We find that, indeed, the exogenous response is largely impervious to task instructions; it simply flips its sign relative to the correct choice, and this largely explains the drastic differences in psychometric performance between the two tasks. Thus, saccadic choices are strongly dictated by the alignment between salience and behavioral goals.

**\*For correspondence:**
esalinas@wakehealth.edu

## Editor's evaluation

When subjects are instructed to produce saccades away from suddenly appearing visual targets under time pressure, early saccades tend to be directed incorrectly to the peripheral target, suggesting that exogenous and endogenous signals that are related to the target position and instruction, respectively, compete to control the motor responses. In this study, the authors provide further evidence for the independence of these two processes by showing that they can account for the temporal evolution of correct saccades regardless of the instruction, stimulus luminance or motor bias.

## Introduction

The oculomotor system of primates specifies a new target to look at every 200–250 ms. Two components of visuospatial attention contribute to the underlying selection process: exogenous mechanisms, which respond to the salience of the objects in view (typically dictated by physical properties such as size, luminance, or motion), and endogenous mechanisms, which prioritize those objects according to their relevance to current behavioral goals (*Itti and Koch, 2001*; *Theeuwes, 2010*; *Wolfe and Horowitz, 2017*).

**Figure 1.** The urgent tasks. (**a**) The compelled antisaccade (CAS) task. After a fixation period (500, 600, or 700 ms), the central fixation point disappears (Go), instructing the participant to look to the left or to the right within 425 ms. The cue is revealed later (Cue on, ±8°), after a time gap that varies unpredictably across trials (Gap, 0–350 ms). The correct response is an eye movement away from the cue, to the diametrically opposite location (Saccade, white arrow). (**b**) The compelled prosaccade (CPS) task. The sequence of events is the same as for the compelled antisaccade task, except that the correct response is an eye movement toward the cue. In all trials, the cue location and gap are selected randomly; the reaction time (RT) is measured between the onset of the go signal and the onset of the saccade; and the raw processing time (rPT) is measured between cue onset and saccade onset (calculated as RT − gap).

The online version of this article includes the following figure supplement(s) for figure 1:

**Figure supplement 1.** Nonurgent variants of the tasks.

The antisaccade task, in which the subject is required to look away from a lone visual cue, is ideal for examining this distinction because it decouples stimulus encoding and response preparation (*Munoz and Everling, 2004*). For a correct antisaccade to take place, endogenous, top-down influences must override the natural tendency to look to the cue. Indeed, in both imaging studies in humans (*Brown et al., 2007*; *Anderson et al., 2008*) and single-neuron recordings in nonhuman primates (*Everling et al., 1998*; *Everling et al., 1999*; *Everling and Munoz, 2000*), antisaccade performance is generally characterized as a conflict between volitional and sensory-driven, reflexive responses. The capacity to resolve such conflict is considered so fundamental, that reaction time (RT) and accuracy measurements in the antisaccade task are commonly used as clinical markers for cognitive dysfunction (*Klein and Foerster, 2001*; *Klein et al., 2003*; *Hutton and Ettinger, 2006*; *Fielding et al., 2009*; *Johansson et al., 2022*).

This work has established key qualitative distinctions between exogenous and endogenous mechanisms. However, resolving their dynamic interactions and specific, real-time contributions to the saccade selection process has been difficult because that requires temporal resolution far greater than can be achieved with traditional psychophysical tasks.

In a recent study (*Salinas et al., 2019*), we used an urgent version of the antisaccade task (*Figure 1a*) to dissociate the exogenous and endogenous contributions to performance with millisecond precision. In the urgent task design (*Stanford and Salinas, 2021*), the go signal that instructs the subject to respond is given before the visual cue that specifies the correct choice (an eye movement away from the cue, in this case). This way, motor plans are initiated early, so participants guess on some trials and make informed choices on others, but critically, each outcome depends fundamentally on how much time the participant has to see the cue before committing to a response, a quantity that we call the raw processing time (rPT). If choice accuracy is then plotted as a function of rPT, the result is a novel psychometric measure — the tachometric curve — that describes the participant's perceptual performance on a moment-by-moment basis (*Stanford and Salinas, 2021*).

In the urgent antisaccade experiment (*Salinas et al., 2019*), the tachometric curve revealed a unique feature: a range of processing times roughly between 90 and 130 ms in which performance plummeted to nearly 0% correct (*Figure 2d*). In other words, there was a range of conditions (i.e., states of motor preparation) indexed by processing time over which the cue onset would almost inevitably capture the saccade and lead to an error. Thereafter, for rPT ≳140 ms, accuracy increased steadily and reached nearly 100% correct.

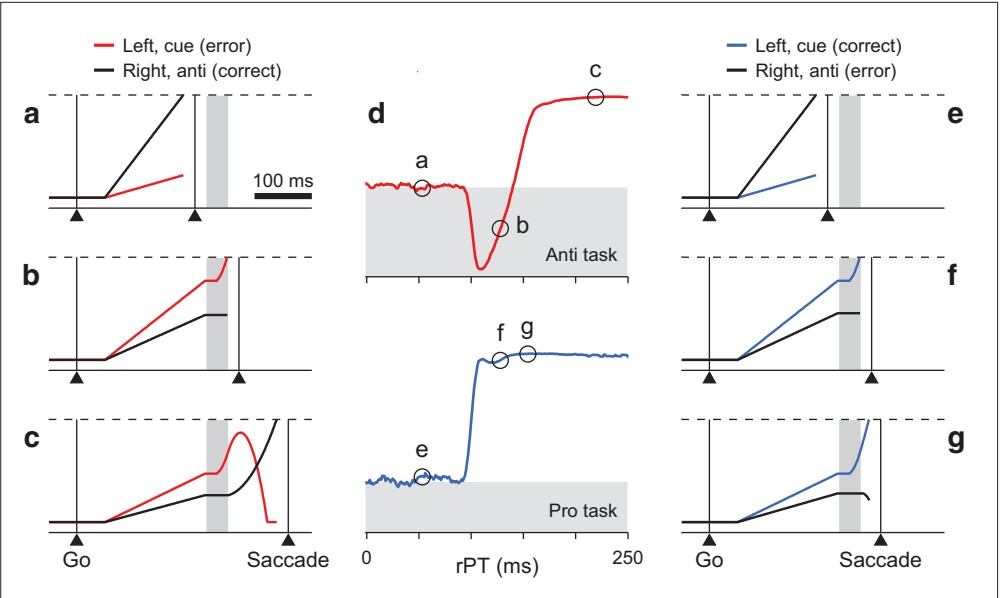

**Figure 2.** Turning a model of antisaccade performance into one of prosaccade performance. (**a–c**) Three single antisaccade trials simulated with the CAS model. The cue is assumed to be on the left and the gap is 150 ms. Traces show motor plans $r_L$ toward the left (red, incorrect) and $r_R$ toward the right (black, correct) as functions of time. During the exogenous response interval (ERI, gray vertical shade), the plan toward the cue accelerates. After the ERI, the incorrect plan decelerates and the correct one accelerates. A saccade is triggered a short efferent delay after activity reaches threshold (dashed lines). Examples include a correct, short-rPT guess (**a**, rPT = 56 ms); an incorrect, captured saccade (**b**, rPT = 133 ms); and a correct, informed choice (**c**, rPT = 219 ms). (**d**) Simulated tachometric curves for the CAS (top, red) and CPS tasks (bottom, blue). The x and y axes correspond to raw processing time and fraction of correct choices, respectively. Gray shades indicate below-chance performance, where chance (white-gray border) is 50% correct. (**e–g**) Three single prosaccade trials with the same initial motor plans as in a–c but simulated with the CPS model. They include an incorrect, short-rPT guess (**e**, rPT = 56 ms); a correct, captured saccade (**f**, rPT = 133 ms); and a correct, informed choice (**g**, rPT = 149 ms). The pro- and antisaccade simulations differed only in the movement that was considered correct, which amounted to swapping the motor plans that were endogenously accelerated and decelerated.

We interpreted these data as the direct manifestation of the conflict between the early attentional pull toward the cue, which is salience-driven, involuntary, and transient, and the later intention to look away, which is task-dependent, voluntary, and sustained. This conflict is hardly detectable during active fixation, but becomes obvious when saccade plans are ongoing. As such, it was instantiated within a saccade competition model in which two opposing motor plans race against each other to trigger an eye movement to the right or to the left (*Figure 2a–c*). As elaborated below, these plans initially advance at randomly drawn rates, but afterward, time permitting, they are steered by perceptual information. The exogenous signal arrives early (~80 ms after cue onset) and rapidly accelerates the plan toward the cue regardless of task instructions; the endogenous signal arrives (~25 ms) later to accelerate the correct plan and decelerate the incorrect one as stipulated by the task. These dynamics replicated all aspects of the behavioral data in great detail.

Here, we leverage this quantitative framework to further test a fundamental conclusion derived from that study: that the exogenous modulation of ongoing motor plans is highly stereotyped and largely independent of behavioral context. We use the antisaccade model to predict the performance that should be observed during an urgent prosaccade task (*Figure 1b*) assuming that this independence hypothesis is correct, and we test the parameter-free predictions in human participants performing both urgent prosaccades and urgent antisaccades. In essence, the exogenous modulation should evolve with the same timecourse regardless of task instructions, and should favor the motor plan toward the cue in the same way whatever the state of development of the competing motor plans.

To anticipate the main findings, the differences between pro- and antisaccade performance are dramatic, but can be largely accounted for by a fixed, exogenously driven response (an exogenous response, for brevity) that is either aligned or misaligned with the endogenously defined goal.

# Results

## A quantitative framework for urgent anti- and prosaccade performance

In our previous study (*Salinas et al., 2019*), we developed a neurophysiologically feasible model that quantitatively replicated the rich psychophysical dataset obtained in the compelled antisaccade task (the CAS model). Now, with a minimal modification, we generate a model of the compelled prosaccade task (the CPS model). The modification corresponds to simply making the top-down, endogenous response favor a saccade to the cue instead of away from the cue — without changing anything else.

The CAS model consists of two variables, $r_L$ and $r_R$, that represent oculomotor activity favoring saccades toward left and right locations (*Figure 2a–c*, red and black traces). The first one of these motor plans to exceed a fixed threshold level (*Figure 2a–c*, dashed lines) triggers a saccade, to the left if $r_L$ crosses the threshold first, or to the right if $r_R$ does. In each trial, $r_L$ and $r_R$ start increasing in response to the go signal, initially advancing with randomly drawn buildup rates. This ramping process may end in a random choice (i.e., a guess; *Figure 2a*) if the gap interval between the go and the cue onset is long and/or one of the initial buildup rates is high enough. Otherwise, time permitting, the arrival of the cue information modifies the ongoing motor plans in two ways.

First, when the cue onset is detected by the oculomotor circuit, the motor competition is biased toward the cue location during a time period that we refer to as the exogenous response interval, or ERI (*Figure 2a–c*, gray vertical shades). Specifically, this means that, during the ERI, the motor plan toward the cue is first briefly halted and then accelerated (*Figure 2b, c*, red traces during gray interval), whereas the motor plan toward the opposite, or 'anti', location is halted throughout (*Figure 2b, c*, black traces during gray interval). These modulations of the ongoing motor activity constitute the involuntary, exogenous response.

Then, after the ERI ends, once the cue location has been interpreted in accordance to task rules as 'opposite to the target', the motor selection process is steered toward the correct choice. Specifically, the (erroneous) plan toward the cue is decelerated (*Figure 2c*, red trace after gray interval) and the (correct) plan away from the cue is accelerated (*Figure 2c*, black trace after gray interval). These modulations informing the correct choice constitute the top-down, endogenous response.

With these elements in place, the CAS model was able to reproduce the full range of psychophysical results obtained in the urgent antisaccade task, both on average and in individual participants (with parameter values adjusted accordingly; *Salinas et al., 2019*). In particular, it quantitatively matched the full RT distributions for correct and error trials (see Figure 8 and its supplements in *Salinas et al., 2019*). Moreover, the model simulations accurately replicated the tachometric curve (*Figure 2d*, top), the function relating choice accuracy to rPT, which is the key behavioral metric in such urgent tasks. In the rest of the paper, we consider the CAS model with parameters fitted to the pooled data from six participants ( see Table 1, high-luminance cue, in *Salinas et al., 2019*). The tachometric curve simulated for this case (*Figure 2d*, top) directly illustrates the conflicting interaction between exogenous and endogenous mechanisms that characterizes the CAS task: When the cue is seen for less than 90 ms or so (as in *Figure 2a*), the saccades are uninformed and the success rate stays near chance. When the cue is seen for more than 140 ms or so (as in *Figure 2c*), most saccades are informed and the success rate rises steadily above chance and toward 100% correct. But in between, performance drops to nearly 0% correct, as a large fraction of saccades are drawn to the wrong location. Those saccades are captured by the cue (as in *Figure 2b*).

To generate predictions about performance in prosaccade trials, we modified the CAS model so that the endogenous signal accelerated the motor plan toward the cue and decelerated the competing plan toward the anti location. That is, we simply swapped the target and distracter designations for the two locations without altering any parameter values. In this way, in the resulting CPS model the exogenous modulation during the ERI still boosts the motor plan toward the cue, just as it did originally (*Figure 2f*, compare to panel b). Crucially, though, those motor plans that are propelled past the threshold, which previously resulted in erroneous captured saccades, now correspond to correct choices. Furthermore, the informed endogenous signal now typically acts to reinforce the motor plan that is already ahead by the end of the ERI (*Figure 2g*, compare to panel c). All this results in a predicted tachometric curve for prosaccade performance for which the success rate grows steadily with rPT, rising very early and very steeply (*Figure 2d*, bottom).

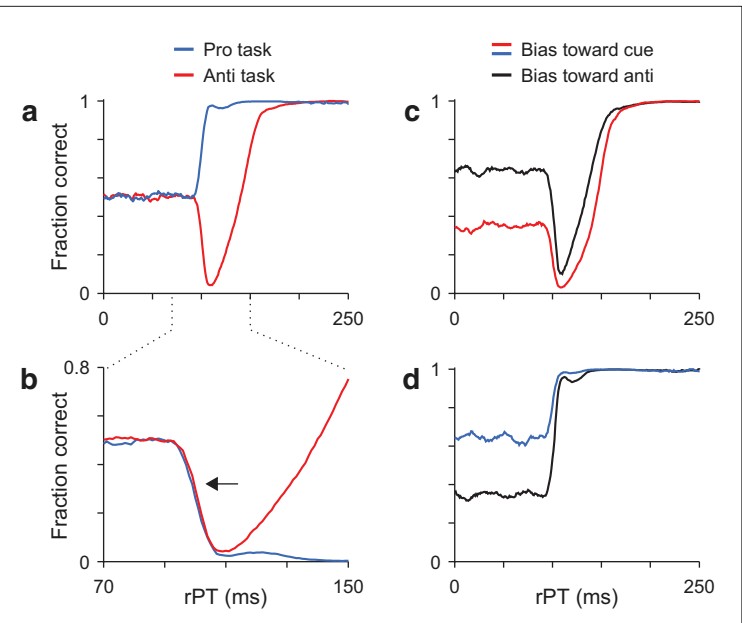

**Figure 3.** Model predictions. (**a**) Simulated tachometric curves for the pro- (blue trace) and antisaccade tasks (red trace). Same curves as in **Figure 2d**, but superimposed. (**b**) Same data as in a, but shown over a smaller rPT range and with the prosaccade curve inverted (blue trace) relative to the chance level. Note that the initial departure from chance follows the same timecourse for the two tasks (arrow). (**c**) Antisaccade tachometric curves conditioned on cue location. Traces are expected results if the participant consistently guesses in one direction and trials are split into two groups: with the cue on the preferred side (red trace) or with the cue on the nonpreferred side (black trace). Note that the rise toward asymptotic performance occurs later when the initial motor bias is in the direction of the cue. (**d**) As in c, but for the prosaccade task. Note that, in this case, the rise in performance is similar regardless of the initial bias.

## Specific model predictions

The fundamental assumption behind the CPS model just described is that the exogenous response is entirely insensitive to behavioral context, so the endogenous signal reinforces the correct choice in exactly the same way in the pro and anti tasks — that is, the acceleration and deceleration terms simply swap their spatial assignments — regardless of the state of development of the motor alternatives. This is one extreme in the spectrum of possible outcomes; opposite to this 'full-independence' scenario is the possibility that the exogenous and endogenous modulations are much different in one task than in the other; and, of course, one or both of these mechanisms could also change modestly across conditions. Hence, the more general question is, to what degree are the exogenous and endogenous mechanisms sensitive to behavioral context? We use the CPS model as a benchmark because it instantiates a conceptually simple hypothesis (independence) to yield parameter-free, quantitative predictions — which, as it turns out, to a first approximation are fairly accurate.

The CAS and CPS models make two specific, clear-cut predictions. The first one is that the upswing in performance during prosaccade trials should follow the same timecourse as the initial downswing in performance during antisaccade trials (**Figure 3a, b**). In other words, the initial departure from chance should exhibit the same dependence on rPT but going in opposite directions, toward 100% versus toward 0% correct. This is a direct consequence of the exogenous response reinforcing the motor plan toward the cue with the same timing and intensity in the two tasks.

The second prediction is about the rise toward asymptotic performance when there is a motor bias, that is, when the participant tends to guess more toward one side than the other. If such a bias exists, then during antisaccade trials in which the cue happens to appear on the preferred guessing side, the rise in success rate is expected to occur late (**Figure 3c**, red trace). In contrast, during antisaccade trials in which the cue happens to appear opposite to the preferred guessing side, the success rate should rise earlier (**Figure 3c**, black trace). This difference arises because the exogenous and endogenous modulations are always opposite, so it is more difficult for the endogenous signal to prevent a

capture when the exogenous response and the motor bias reinforce each other. On the other hand, during prosaccade trials the rise in success rate is expected to be very similar regardless of where the cue appears relative to the guessing direction (*Figure 3d*). In this case, the exogenous and endogenous modulations are aligned and can easily overcome any bias in the initial motor plans, so the rise in success rate is always very steep.

These two predictions are complementary in that they probe bottom-up and top-down components of the selection process. The first one tests whether the sensory-driven exogenous response itself changes across tasks, whereas the second one tests whether the endogenously driven recovery depends on the state of development of the competing motor alternatives.

## The exogenous response is highly insensitive to behavioral context

To test the model predictions, 10 participants (6 female, 4 male) were recruited. The experimental procedures were largely the same as in the earlier report (*Salinas et al., 2019*) and are summarized in Materials and methods. Any differences in approach are described there, along with analysis methods specific to this study.

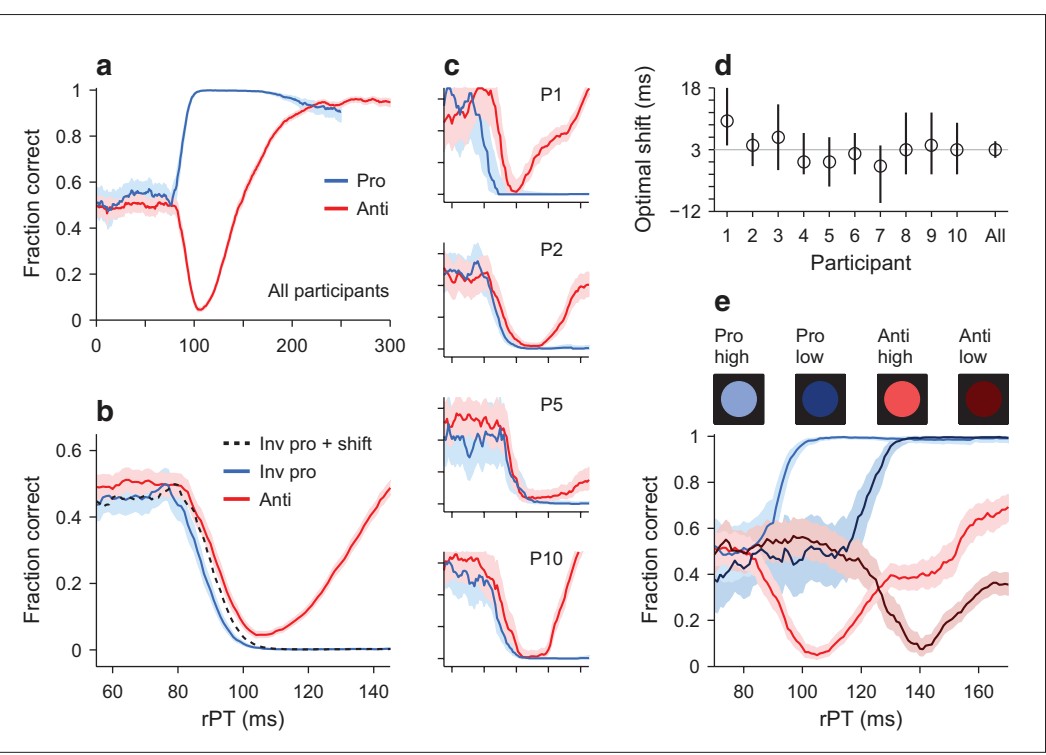

**Figure 4.** Early departure from chance during pro- and antisaccade performance. (**a**) Tachometric curves for pro- (blue trace, 24,638 trials) and antisaccade trials (red trace, 22,608 trials) combined over 10 participants. Shades indicate 95% confidence intervals (CIs) across trials. (**b**) Same data as in a, but shown over a smaller rPT range and with the prosaccade curve inverted (blue trace) relative to the chance level. The dotted line is the inverted prosaccade curve shifted by 3 ms along the x-axis. (**c**) Pro- and antisaccade curves for four individual participants. Same x and y axes as in b, and same format. (**d**) Optimal shift for each participant. The optimal shift (along the rPT axis) minimized the difference between the inverted pro curve and the anti curve for rPTs in the [55, 105] ms interval. The gray line marks the optimal shift (3 ms) for the pooled data in b. Error bars are 95% CIs. (**e**) Tachometric curves pooled from three participants tested with both the standard high-luminance cues (bright colors) and low-luminance cues (dark colors). The rPT range is truncated to better appreciate the luminance-driven right shift, which is similar for the pro and anti curves (for full range, see *Figure 4—figure supplement 3*).

The online version of this article includes the following figure supplement(s) for figure 4:

**Figure supplement 1.** Performance in blocked versus interleaved trials.

**Figure supplement 2.** Perceptual performance is largely invariant with respect to gap value.

**Figure supplement 3.** Individual results for the three participants who performed the luminance experiment.

All the participants performed pro and anti trials in two ways: in blocks of same-task trials, and in blocks of randomly interleaved pro- and antisaccades (Materials and methods). The experiment was designed this way because it seemed possible that the two conditions could lead to major differences in performance, via task-switching costs, for instance (*Wylie and Allport, 2000*; *Monsell, 2003*). However, the observed differences were, in fact, very small (*Figure 4—figure supplement 1*) and had no bearing on any of the subsequent analyses. Thus, in what follows, we consider all the trials of each task aggregated across blocked and interleaved conditions.

For the compelled antisaccade task, the pooled tachometric curve (*Figure 4a*, red trace), generated by combining the data across participants, replicated our earlier result (*Salinas et al., 2019*). It exhibited the characteristic drop in success rate due to potent oculomotor capture in the range between 90 and 140 ms, approximately, with the subsequent recovery reaching nearly 100% correct after ~200 ms of cue viewing time. More importantly, for the compelled prosaccade task the shape of the pooled tachometric curve (*Figure 4a*, blue trace) was as predicted by the model. In this case, the success rate rose steadily, early, and very steeply.

We stress that the tachometric curve is the main metric in urgent tasks because it characterizes performance independently of RT (*Stanford et al., 2010*; *Salinas et al., 2019*; *Stanford and Salinas, 2021*; *Seideman et al., 2022*) or gap (*Figure 4—figure supplement 2*). These variables explain very little once rPT has been taken into account.

To evaluate the first prediction quantitatively, we compared the downswing in antisaccade performance to the upswing in prosaccade performance (*Figure 4b*). First, the pro curve was inverted so that it decreased monotonically, and then a shift analysis was carried out to determine the optimal shift along the rPT axis that would maximize the overlap between the two curves (Materials and methods). Whereas according to the prediction the optimal time shift should be zero (*Figure 3b*, arrow), experimentally, we found that the optimal shift was, on average, 3 ms within a 95% confidence interval (CI) of [1, 5] ms (*Figure 4b, d*). That is, in prosaccade trials, the effect of the exogenous response manifested about 3 ms earlier than in antisaccade trials. This time difference was of comparable magnitude across individual participants (*Figure 4c*), which had optimal shifts in the [−1, 10] ms range (*Figure 4d*). Also, the result was very similar when green and magenta cues were assigned to pro and anti trials, respectively (participants 1–5; Materials and methods), and when the color assignments were flipped (participants 6–10). In general, the upswing in prosaccade performance had just a slightly earlier onset than the downswing in antisaccade performance.

To confirm these results, we ran a variant of the experiment in which the cue could be of high or low luminance (Materials and methods). We knew that, in the antisaccade task, less luminant cues produce capture at longer processing times (*Salinas et al., 2019*), so we reasoned that the rise in prosaccade performance should demonstrate the same temporal dependence — if it is driven by the same exogenous signal. The low luminance level was chosen to elicit a robust rightward shift of the anti tachometric curve (*Salinas et al., 2019*), and the high-luminance level was identical to that in the main experiment. In this case, the cue luminance (high or low) and task type (pro or anti) were randomly and independently chosen in each trial, and data were collected from three participants.

As expected, in antisaccade trials the decrease in cue luminance resulted in a rightward shift of the tachometric curve (*Figure 4e*, red curves), which was of 35 ms ([33, 38] ms 95% CI), as determined by an optimal-shift analysis. And critically, a comparable delay was observed in prosaccade trials (*Figure 4e*, blue curves), for which the lower luminance resulted in a shift of 30 ms ([26, 34] ms 95% CI). The results were qualitatively the same for the three participants (*Figure 4—figure supplement 3*). In conclusion, the data indicate that, although they are not quite identical, the exogenous signal that rapidly propels prosaccade performance above chance must be very similar to the one that drives antisaccade performance below chance.

## Visuomotor dynamics revealed by motor biases

The second prediction derived from the model applies when participants exhibit a motor bias, that is, a preference for looking to one side more than the other. Fortuitously, we did not have to manipulate the experimental conditions to induce such a preference because participants adopted one spontaneously.

When we analyzed our participants' performance as a function of their prior choices, we noticed a trend: their guesses in a given trial were generally toward the location where they should have

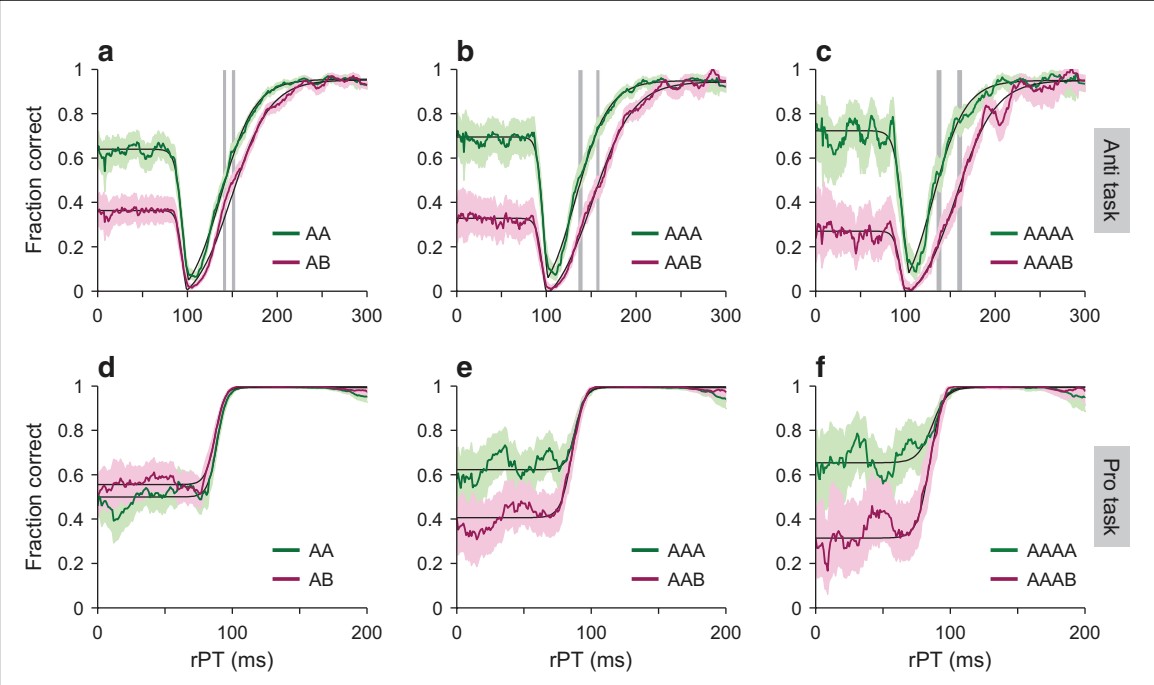

**Figure 5.** Tachometric curves conditioned on target-location history. Each panel shows choice accuracy as a function of processing time when the target location in a given trial was the same as in previous trials (green curves) or when it was different (magenta curves). Note that the participants' guesses (rPT ≤80 ms) tended to be toward the previous target location. (**a–c**) Performance in the compelled antisaccade task conditioned on the history of prior target locations going back 1, 2, or 3 trials (panels **a, b, c**, respectively). A and B labels stand for left or right target locations, and indicate patterns of repeats (AA, AAA, AAAA) or switches in location (AB, AAB, AAAB) preceding each choice. Shaded error bands indicate 95% CIs across trials. Continuous black curves are fits to the empirical data. Gray vertical shades indicate 95% CIs for the curve rise points. (**d–f**) As in **a–c**, but for the compelled prosaccade task. Results are for data pooled across participants.

The online version of this article includes the following figure supplement(s) for figure 5:

**Figure supplement 1.** Effect of motor bias on antisaccade performance in individual participants.

responded in the previous trial (i.e., the prior target location). The psychometric manifestation of this tendency is a systematic deviation from chance observed when the trials are sorted according to the history of prior target locations. Specifically, the tachometric curve conditioned on a repeated target location (*Figure 5a*, green trace) demonstrated a positive offset (above chance) in the range of processing times for which participants must guess (rPT ≤80 ms), whereas the tachometric curve conditioned on a switch in target location (*Figure 5a*, magenta trace) demonstrated a negative offset (below chance) in the same range. This effect was cumulative; the more target repetitions, the larger the offsets (*Figure 5*, compare across columns). It was also observed in most participants (*Figure 5— figure supplement 1*) and in both pro and anti trials.

Given this motor bias, we analyzed the tachometric curves conditioned on prior target location and measured the rise point in each case; for anti trials, this is the rPT at which the success rate is halfway between minimum and asymptotic (*Figure 5a–c*, gray vertical shades), and for pro trials, it is the rPT at which the success rate is halfway between chance and asymptotic (Materials and methods). We found that both aspects of the second model prediction were correct. In the antisaccade task, the rise toward asymptotic performance occurred late when the motor bias was aligned with the cue location (*Figure 5a–c*, magenta curves) and early when it was aligned with the anti, or target location (*Figure 5a–c*, green curves); whereas in the prosaccade task, the rise was essentially the same in the two cases (*Figure 5d–f*). It is notable that the effect in anti trials grew stronger as the bias became more extreme, and it is instructive to calculate by how much.

Consider the tachometric curves conditioned on one previous trial (*Figure 5a*). As the chance level went from 0.36 ([0.34, 0.39] 95% CI, AB trials) to 0.64 ([0.61, 0.66], AA trials), the rise point of the tachometric curve went from 154 ms ([149, 154] CI, AB trials) to 143 ms ([139, 144], AA trials). Taking the ratio of the differences gives a drop of approximately 39 ms when traversing the full chance range

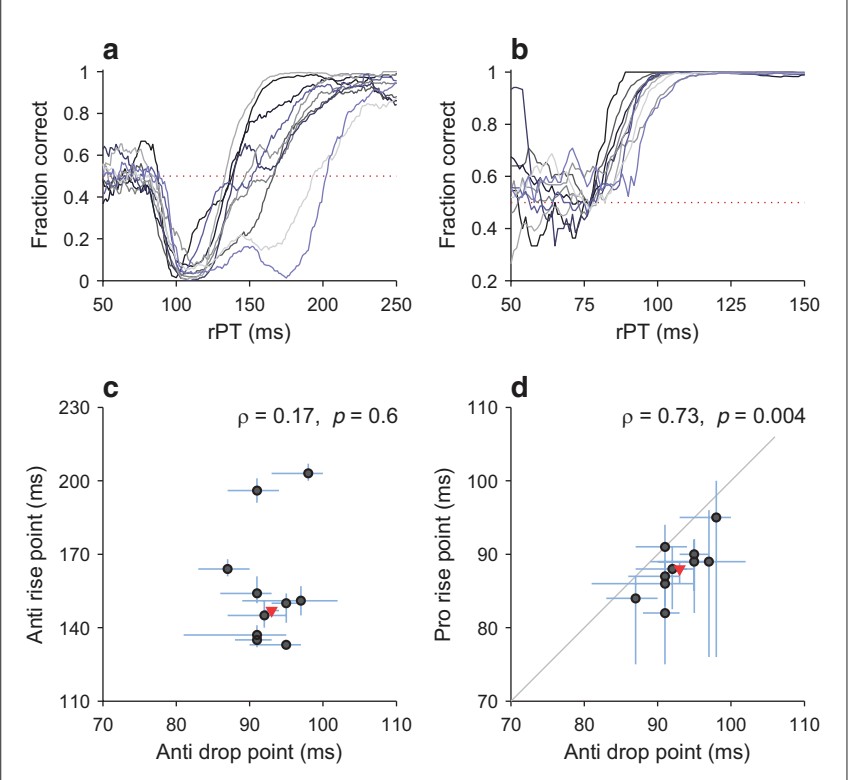

**Figure 6.** Variations in the timing of exogenous and endogenous modulation across participants. (**a**) Tachometric curves in the compelled antisaccade task for all 10 participants. Dotted line indicates chance performance (0.5). (**b**) As in a, but for the compelled prosaccade task. (**c**) Comparison between drop point (*x*-axis; time of early, exogenously driven drop in performance) and rise point (*y*-axis; time of later, endogenously driven rise in performance) in antisaccade trials. Each circle represents one participant. Lines indicate 95% CIs (from bootstrap). The Pearson correlation of the data (*ρ*) and its significance (from permutation test) are indicated. The red triangle marks data from the pooled tachometric curve. (**d**) Comparison between drop point in antisaccade trials (*x*-axis; time of early, exogenously driven drop in performance) and rise point in prosaccade trials (*y*-axis; time of early rise in performance). Same format as in c). Note the positive correlation between the early responses in the two tasks.

from 0 to 1; or equivalently, if the chance level increases by 0.1, the rise point of the curve should drop by about 3.9 ms. The reason why this number is of interest is that it can also be computed for the simulated data. To do this, we simply ran the model (as in *Figure 2c*) with different amounts of motor bias, and for each run we tracked the chance level and the rise point of the simulated tachometric curve (Materials and methods). According to this procedure, the predicted change in rise point over the full chance range was 40 ms ([36, 44] 95% CI; *Figure 5—figure supplement 1b*, blue line and circles), in excellent agreement with the experimental data. Although in this analysis there was considerable variance across individual participants (*Figure 5—figure supplement 1*), the model prediction was very much in line with the data pooled across participants (*Figure 5—figure supplement 1b*, compare triangles and blue circles).

## Consistency in the strength of exogenous capture across participants

As expected, in the two tasks there were noticeable differences in performance between individual participants, and analysis of this variance provided further support for the two hypotheses underlying our model predictions.

For the antisaccade task, the tachometric curves of the participants were quite similar in their initial departure from chance, as saccades were increasingly captured by the cue, but they were much more variable during the later rise in success rate (*Figure 6a*). Across the individual curves, the drop point (the rPT for which the success rate is halfway between chance and the minimum; Materials and methods) ranged from 87 to 98 ms, whereas the rise point ranged from 133 to 203 ms.

More importantly, the values for the drop point and the rise point were not significantly correlated (*Figure 6c*; $p = 0.6$, permutation test), consistent with the idea that, in each subject, the exogenous response that determines the former is independent of the endogenous process responsible for the latter.

For the prosaccade task, the rise point covered a range from 82 to 95 ms across participants (*Figure 6b*), similar to that of the drop point in the anti task. More importantly, however, these two sets of values were significantly correlated (*Figure 6d*; $\rho = 0.73$, $p = 0.004$, permutation test). An alternate analysis based on optimal shifts instead of rise and drop points produced nearly identical correlation numbers. In summary, then, participants that showed an earlier departure from chance in one task also showed an earlier departure from chance in the other, consistent with the notion that, for each individual, the exogenous capture is fundamentally the same in the two tasks.

# Discussion

We used the contrast between urgent pro- and antisaccade performance to test the idea that exogenous and endogenous contributions to saccadic target selection are largely independent. The cue onset and the endogenously defined goal were spatially congruent in one task but incongruent in the other, and yet, in both cases we observed a similarly steep increase in the frequency of saccades made toward the cue. As a result, there were striking differences in the proportion and timing of the correct responses across conditions (pro versus anti), but this was very much as predicted by our race model.

Dependencies between exogenous and endogenous signals could have plausibly manifested in many ways. The exogenous response in anti trials could have been visibly attenuated, delayed, or less sensitive to luminance relative to that in pro trials (*Figure 4*); the exogenous capture in anti trials could have been stronger in interleaved as compared to blocked conditions (*Figure 4—figure supplement 1*); or it could have been much weaker than predicted by the model when the ongoing motor activity pointed away from the cue rather than toward it (*Figure 5*). A substantial difference in any of these cases would have meant that the exogenous response was subject to internal modulation. Discrepancies were, in fact, very slight, largely consistent with the independence assumption — but this was by no means a foregone conclusion. This is not to say that robust modulation of the exogenous response is impossible, but the results circumscribe considerably the conditions under which it may occur.

## Exogenous capture connects numerous visuomotor phenomena

Given our modeling framework, the mechanistic intuition behind these results is straightforward: when saccade-related neural activity is ramping up, the cue onset can accelerate that activity past the threshold beyond which a saccade is inevitably produced — before top-down information can influence the choice. When the cue itself is not the target, as in the antisaccade task, this leads to so-called captured saccades, which are errors. But when the cue itself is the target, as in the prosaccade task, these same 'captured' saccades are correct. In either case, the eye movement is elicited very rapidly, after 75–100 ms of viewing time. The quantitative variations in this process across conditions were tiny, consistent with the hypothesis that the resulting bias on the motor selection process is associated with the detection of the cue and is largely independent of its behavioral relevance or meaning. Perhaps an appropriate descriptor for this effect, as it is thought to occur in oculomotor circuits, is 'exogenous capture'.

Understood this way — as an involuntary and stereotypical bias on motor planning that occurs when a stimulus is detected by oculomotor circuits — exogenous capture potentially encompasses a wide range of oculomotor phenomena (*Salinas and Stanford, 2021*). These include (1) oculomotor capture itself, which is observed when saccades are overtly directed toward salient distracters (*Theeuwes et al., 1998*; *Theeuwes et al., 1999*); (2) express saccades, which are saccades triggered with extremely short latencies (*Kalesnykas and Hallett, 1987*; *Paré and Munoz, 1996*; *Sparks et al., 2000*); (3) exogenous attention, which improves perceptual performance at explicitly cued locations (*Carrasco, 2011*) and has neural correlates that are highly consistent with our framework (*Bisley and Goldberg, 2003*; *Busse et al., 2008*; *Herrington and Assad, 2010*); and (4) attentional capture, which is typically observed when saccades to a target take longer in the presence of salient distracters (*Ruz and Lupiáñez, 2002*; *Theeuwes, 2010*).

Exogenous capture is a potentially useful construct because it represents a specific hypothesis about the neural events that determine, or at least contribute significantly to, all of these behaviors. For example, the variations in RT that define attentional capture can be easily explained by our framework (*Oor et al., 2021*; *Salinas and Stanford, 2021*). In essence, when the distracter-driven acceleration of the cue-aligned plan is relatively weak (weaker than in *Figure 2c*), most saccades are ultimately correct, but their onset is still delayed relative to a control condition with a less salient or absent distracter. That is, there is reason to think that in attentional capture paradigms the exogenous bias is qualitatively identical to that in the model (*Figure 2b, c, f, g*) but weaker, so its overt manifestation (in the RT distribution) is more subtle.

A key consideration is that, during active fixation, motor plans are strongly suppressed, so establishing a quantitative link between visual transients, motor activity, and probability of success in a task is more difficult than under urgent conditions. When motor plans are already underway, exogenous capture becomes strong and highly reliable, and its impact on performance easier to discern across tasks (*Salinas et al., 2019*; *Oor et al., 2021*; *Salinas and Stanford, 2021*).

Work by *Poth, 2021* showed that robust capture phenomena can be generated in urgent tasks that use button presses instead of saccades and impose very different cognitive demands on the choice process. Task conditions akin to pro and anti were distinguished not by different spatial rules, but rather by contextual features that could be congruent (e.g., >>>>>) or incongruent with a central stimulus (e.g., <<><<). The qualitative similarities with our pro- and antisaccade tasks are remarkable, and suggest two critical questions to pursue more broadly: how some visuomotor associations become more reflexive than others, and how established reflexive tendencies are overcome.

## No clear target for inhibitory control

Success in the antisaccade task is typically conceptualized as the result of inhibitory control. That is, it is assumed that an inhibitory process suppresses the reflexive reaction of looking at the suddenly appearing visual cue (*Munoz and Everling, 2004*; *Wiecki and Frank, 2013*; *Coe and Munoz, 2017*). But, what exactly is supposed to be inhibited?

Several factors determine success in an antisaccade trial. On average, weaker preparatory activity, due to a late break of fixation or to low urgency to respond, leads to lower probability of capture, consistent with single-neuron recordings in the superior colliculus and the frontal eye field (*Everling et al., 1999*; *Everling and Munoz, 2000*). A developing motor plan that, by chance, is spatially incongruent with the cue is less likely to be captured than one that is congruent (*Figure 5a–c* and *Figure 5—figure supplement 1b*), also in agreement with neural recordings (*Everling et al., 1998*; *Everling et al., 1999*; *Everling and Munoz, 2000*). The intensity of the exogenous response is certainly critical, with less salient cues leading to later and less pronounced capture (*Figure 4e* and *Figure 4—figure supplement 3*; *Salinas et al., 2019*). But as far as we can tell, there is no mechanism for preventing the exogenous burst; at most it may be slightly modulated by task conditions (*Figure 4* and *Figure 4—figure supplement 1*; *Everling et al., 1998*; *Everling and Munoz, 2000*; see also *Gu et al., 2016*; *Salinas and Stanford, 2018*, *Salinas et al., 2019*; *Buonocore et al., 2017*; *Bompas et al., 2020*; *Salinas and Stanford, 2021*). Finally, the probability of a successful antisaccade is higher when the endogenous modulation that steers the motor plans is stronger or arrives earlier. This signal does implicate inhibition because it must decelerate the incorrect motor plan, but according to modeling results, this applies equally to any urgent saccadic choice that is perceptually informed (*Stanford et al., 2010*; *Shankar et al., 2011*; *Costello et al., 2013*; *Salinas and Stanford, 2013*; *Salinas and Stanford, 2018*; *Salinas and Stanford, 2021*; *Seideman et al., 2018*; *Stanford and Salinas, 2021*), including prosaccades. Furthermore, endogenous inhibition does not act alone, it complements endogenous excitation that is necessary for accelerating the correct motor plan.

All this indicates that the role of inhibition in antisaccade performance is likely widespread and quite nuanced (see *Aron, 2007*). In summary, then, we found that the capture mechanism is the same in the urgent pro- and antisaccade tasks, and that it is the consequence of a robust exogenous burst of oculomotor activity that always occurs. Therefore, it remains to be determined whether antisaccade performance depends critically on a specific circuit element being inhibited, but the obvious candidate, the exogenous response itself, is not it.

## Materials and methods

In general, methods were the same as in the preceding study (*Salinas et al., 2019*). Here, we summarize key elements of the approach and detail new procedures that were implemented specifically for this experiment.

### Subjects and setup

Experimental subjects were 10 healthy human volunteers, 4 male and 6 female, with a median age of 28 and ranging between 22 and 63 years. They were recruited from the Wake Forest School of Medicine and Wake Forest University communities. All had normal or corrected-to-normal vision. All participants provided informed written consent before the experiment. All experimental procedures were conducted with the approval of the Institutional Review Board of Wake Forest School of Medicine (IRB00035241).

As in the preceding study (*Salinas et al., 2019*), the experiment took place in a semi-dark room. The participants sat on an adjustable chair, with their chin and forehead supported, and viewed stimuli presented on a VIEWPixx LED monitor (VPixx Technologies Inc, Saint Bruno, Quebec, Canada; 1920 × 1200 screen resolution, 120 Hz refresh rate, 12-bit color) at a distance of 57 cm. Eye position was recorded using an EyeLink 1000 infrared camera and tracking system (SR Research, Ottawa, Canada; 1000 Hz sampling rate). For this experiment, stimulus presentation and data collection were controlled using the Psychtoolbox 3.0 package (*Brainard, 1997*; *Kleiner et al., 2007*) and custom Matlab scripts (*Goldstein et al., 2022*).

### Behavioral tasks

The sequence of events was the same in the pro- and antisaccade tasks, and is described in *Figure 1*. As in the earlier study (*Salinas et al., 2019*), the gap values used were −200, −100, 0, 75, 100, 125, 150, 175, 200, 250, and 350 ms, where negative numbers correspond to delays in nonurgent versions of the task (*Figure 1—figure supplement 1*); that is, compelled and easy, nonurgent trials were interleaved. In each trial, the gap value and cue location (−8° or 8°) were randomly sampled. The cue was a circle of 1.5° diameter. Auditory feedback (a beep) was provided at the end of the trial if the saccadic choice was made within the allowed RT window (425 ms); no sound was played if the limit was exceeded. No feedback was given about the correctness of the spatial choice. The intertrial interval was 350 ms.

Data were collected in blocks of 150 trials, with 2–3 min of rest allowed between blocks. For the main experiment, each participant completed 9 or 10 blocks of prosaccade trials, 9 or 10 blocks of antisaccade trials, and 18 blocks of pro and anti trials interleaved in equal proportions. For half of the participants (P1–P5), the cue was green (RGB vector [0 0.88 0]) in prosaccade trials and magenta (RGB vector [1 0.32 1]) in antisaccade trials. For the other half (P6–P10), the color assignments were swapped. The green and magenta stimuli had the same luminance (48 cd m$^{-2}$), as determined by a spectrophotometer (i1 Pro 2 from X-Rite, Inc, Grand Rapids, MI). In blocks in which pro and anti trials were interleaved, the color of the fixation point was the same as that of the cue and indicated to the participant the type of trial that was coming up. In blocks of single-task trials, the fixation point was gray (RGB vector [0.25 0.25 0.25]).

### Secondary experiment

Additional data were collected in a secondary experiment in which luminance varied across trials (*Figure 4e*). One of the previous volunteers and two newly recruited ones participated (all female). The sequence of events in the tasks was exactly the same as in the main experiment. In this case, however, in each trial, the task type (pro or anti) and cue luminance (high or low) were independently sampled, so the four possible combinations were randomly interleaved. The color of the fixation spot, which was always the same as that of the upcoming cue, indicated the correct response: green stimuli corresponded to prosaccades and magenta stimuli to antisaccades. The green and magenta cues of high luminance (48 cd m$^{-2}$) were the same as in the main experiment. The green (RGB vector [0 0.0733 0]) and magenta (RGB vector [0.0833 0.0267 0.833]) cues of low luminance (0.24 cd m$^{-2}$) were chosen so that significant rightward shifts of the tachometric curve would be generated (*Salinas et al., 2019*).

## Data analysis

All results are based on the analysis of urgent trials (gap ≥0) only; easy, nonurgent trials (gap <0) were excluded. All data analyses were carried out using customized scripts written in Matlab (The MathWorks, Natick, MA), as detailed previously (*Salinas et al., 2019*).

The RT was always measured as the time elapsed between the go signal (fixation offset) and saccade onset (equal to the first time point after the go for which the eye velocity exceeded a threshold of 40°/s). Trials were scored as correct or incorrect based on the direction of the first saccade made after the go signal. Trials with fixation breaks (aborts) or primarily vertical saccades were excluded from analysis. Otherwise, completed trials were scored and included even if they exceeded the allotted time limit.

Processing times were computed as rPT = RT – gap, where all quantities are specific to each trial. To generate tachometric curves, trials were grouped into rPT bins that were shifted every 1 ms, and the numbers of correct and incorrect responses were tallied for each bin. From these, we calculated the fraction of correct responses in each bin and, using binomial statistics, CIs for it. The bin size was 21 ms in all cases except *Figure 4—figure supplement 3*, where it was set to 41 ms.

To quantify perceptual performance, each tachometric curve was fitted with a continuous analytical function. The fits served to determine key characteristic metrics from the empirical curves, which are inherently noisy. For the prosaccade task, the fitting curve was an increasing sigmoid

$$s(x) = B + \frac{A - B}{1 + \exp(-\frac{x - C}{D})} \tag{1}$$

where $C$ is the curve rise point, that is, the rPT at which the fraction correct is halfway between chance (equal to $B$) and asymptotic (equal to $A$). For the antisaccade task, the fitting curve was defined as

$$v(x) = \max(s_L(x), s_R(x), 0) \tag{2}$$

where the maximum function, $\max(a, b, c)$, returns the largest of $a$, $b$, or $c$. Also, $s_L$ and $s_R$ are two sigmoidal curves given by

$$s_L(x) = B_{LR} + \frac{A_L - B_{LR}}{1 + \exp(\frac{x - C_L}{D_L})} \tag{3}$$

$$s_R(x) = B_{LR} + \frac{A_R - B_{LR}}{1 + \exp(\frac{x - C_R}{D_R})} \tag{4}$$

where $s_L$ tracks the left (decreasing) side of the tachometric curve and $s_R$ tracks the right (increasing) side. In this case, $C_L$ is the curve drop point, that is, the rPT at which the fraction correct is halfway between chance (equal to $A_L$) and minimum (equal to $B_{LR}$), whereas $C_R$ is the curve rise point, which in this case corresponds to the rPT at which the fraction correct is halfway between the minimum ($B_{LR}$) and asymptotic (equal to $A_R$). Previously (*Salinas et al., 2019*), we referred to the antisaccade rise point as the curve centerpoint, but the new nomenclature is more clear and more suitable for comparing results between pro and anti conditions.

For any given tachometric curve, parameter values ($A$, $B$, $C$, etc.) were found that minimized the mean absolute error between the experimental data and the analytical curves using the Matlab function `fminsearch`. In most cases, chance levels were fixed at 50% correct by setting $B = 0.5$ and $A_L = 0.5$, so these parameters did not vary during the minimization. However, when analyzing motor biases (*Figure 5*), these parameters were allowed to vary.

In this study, we focus on the drop point $C_L$ and the rise points $C$ and $C_R$. CIs for these quantities were obtained by bootstrapping (*Davison and Hinkley, 2006*; *Hesterberg, 2014*), as done previously (*Salinas et al., 2019*). This involved resampling the trial-wise data with replacement, refitting the resulting (resampled) tachometric curves, storing the new parameter values, and repeating the process many times to generate parameter distributions. Reported 95% CIs correspond to the 2.5 and 97.5 percentiles obtained from the bootstrapped distributions based on 1000–10,000 iterations.

In addition to comparisons between $C$, $C_L$, and $C_R$ values, to evaluate differences in timing between two tachometric curves (as in *Figure 4b–d*), we also performed a time-shift analysis, which was as follows. First, one curve, $f_1(x)$, was designated as the reference and remained fixed. Then, transformed versions of the second curve, $f_2(x)$, were considered that would minimize the absolute difference between it and the reference curve over a range of $x$ values (i.e., rPTs). Three transforming

operations were considered: a shift along the *y*-axis (i.e., a change in baseline), a change in gain, and a shift along the *x*-axis. In practice, values of the parameters $\Delta b$, $g$, and $\Delta x$ were found that minimized the error

$$E = \left\langle \, \left| \, g \left( \Delta b + f_2(x + \Delta x) \right) - f_1(x) \, \right| \, \right\rangle \tag{5}$$

where the angle brackets indicate an average over all rPT values in the specified range. For identical tachometric curves $f_1$ and $f_2$ the optimal parameters would be $\Delta b = 0$, $g = 1$, and $\Delta x = 0$. Although the quantity of interest is the relative time shift between the curves, $\Delta x$, the gain factor and change in baseline are included to account for the fact that the two curves may saturate at somewhat different levels, or may be slightly offset in the vertical direction from each other, which would tend to overestimate the magnitude of the optimal shift, that is, the $\Delta x$ for which the error is minimized. Notably, the effect of $\Delta b$ and $g$ was modest; results varied by approximately 2 ms when the procedure allowed variations in $\Delta x$ only, in which case $\Delta b$ and $g$ were fixed at 0 and 1, respectively.

This shift analysis applies to curves $f_1$ and $f_2$ that have similar shapes. To compare the early departure from chance in pro- versus antisaccade curves, the prosaccade curve was first inverted (*Figure 4b, c*) relative to the chance level by applying the transformation $f(x) \rightarrow 1 - f(x)$. Then the above procedure for determining $\Delta x$ was carried out. CIs for $\Delta x$ were again obtained by bootstrapping, that is, by resampling the data with replacement, recalculating the tachometric curves $f_1$ and $f_2$, and rerunning the entire minimization process. In this way, by iterating this resampling procedure many times, distributions of $\Delta x$ values were generated.

## Model simulations

Predictions were made with the CAS and CPS models as explained in Results. The CAS model is the exact same accelerated race-to-threshold model developed previously (for parameter values, see Table 1 in *Salinas et al., 2019*, high-luminance cue condition). The CAS model thus represents typical, or average, urgent antisaccade performance, as it replicates the psychometric results obtained by pooling all the trials from the six subjects that participated in the prior experiment. The Matlab scripts for simulating the CAS and CPS models differed by a single sign that determined whether the target location was equal to the cue location or opposite to it. Thus, for a given set of parameters, the exogenous and endogenous modulations produced by the two models were the same; the only thing that varied was which plan was accelerated and which decelerated endogenously. To generate predictions, the CPS model was run with the same parameter values as the CAS model.

In standard model simulations with no motor bias, the two initial buildup rates for the competing motor plans were drawn from a bivariate distribution and were assigned randomly to the cue and anti motor plans. Model runs in which a motor bias was included proceeded in the same way, with initial buildup rates drawn from the same bivariate distribution, but these were assigned to the cue and the anti plans with a biased probability. For example, to generate a bias of 0.1 in favor of the cue location, in each simulated trial the larger of the two buildup rates was assigned to the cue side with a 60% probability. Biases of arbitrary magnitude favoring one direction or the other were generated this way.

## Acknowledgements
We thank Denise Anderson for technical and logistical assistance.

## Additional information

### Competing interests
Emilio Salinas: Reviewing editor, eLife. The other authors declare that no competing interests exist.

### Funding

| Funder | Grant reference number | Author |
| --- | --- | --- |
| National Eye Institute | R01EY025172 | Terrence R Stanford<br>Emilio Salinas |

| Funder | Grant reference number | Author |
|---|---|---|
| National Institute of Mental Health | R21MH120784 | Terrence R Stanford Emilio Salinas |
| National Institute of Neurological Disorders and Stroke | T32NS073553 | Allison T Goldstein |

The funders had no role in study design, data collection, and interpretation, or the decision to submit the work for publication.

### Author contributions
Allison T Goldstein, Data curation, Software, Formal analysis, Investigation, Writing – review and editing; Terrence R Stanford, Conceptualization, Data curation, Formal analysis, Supervision, Funding acquisition, Writing – original draft, Project administration, Writing – review and editing; Emilio Salinas, Conceptualization, Data curation, Software, Formal analysis, Supervision, Funding acquisition, Writing – original draft, Project administration, Writing – review and editing

### Author ORCIDs
Allison T Goldstein (ID) http://orcid.org/0000-0002-5475-5965
Terrence R Stanford (ID) http://orcid.org/0000-0003-0759-5599
Emilio Salinas (ID) http://orcid.org/0000-0001-7411-5693

### Ethics
All participants provided informed written consent before the experiment. All experimental procedures were conducted with the approval of the Institutional Review Board of Wake Forest School of Medicine (IRB00035241).

### Decision letter and Author response
Decision letter https://doi.org/10.7554/eLife.76964.sa1
Author response https://doi.org/10.7554/eLife.76964.sa2

## Additional files

### Supplementary files
• Transparent reporting form

### Data availability
The trial-by-trial behavioral data that support the findings of this study are publicly available from Zenodo, https://doi.org/10.5281/zenodo.6757621. Matlab scripts for reproducing analysis results are included as part of the shared data package.

The following dataset was generated:

| Author(s) | Year | Dataset title | Dataset URL | Database and Identifier |
|---|---|---|---|---|
| Goldstein AT, Stanford TR, Salinas E | 2022 | Dataset: Exogenous capture accounts for fundamental differences between prosaccade and antisaccade performance | https://doi.org/10.5281/zenodo.6757621 | Zenodo, 10.5281/zenodo.6757621 |

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
