## [Editor Report]

When subjects are instructed to produce saccades away from suddenly appearing visual targets under time pressure, early saccades tend to be directed incorrectly to the peripheral target, suggesting that exogenous and endogenous signals that are related to the target position and instruction, respectively, compete to control the motor responses. In this study, the authors provide further evidence for the independence of these two processes by showing that they can account for the temporal evolution of correct saccades regardless of the instruction, stimulus luminance or motor bias.

---

## [Decision Letter]

**Decision letter after peer review:**

Thank you for submitting your article "Exogenous capture accounts for fundamental differences between prosaccade and antisaccade performance" for consideration by *eLife*. Your article has been reviewed by 3 peer reviewers, including Daeyeol Lee as Reviewing Editor and Reviewer #1, and the evaluation has been overseen by Timothy Behrens as the Senior Editor.

Essential revisions:

(1) For the trials with low luminance, the asymptotic level of accuracy remained quite low (Figure 4e). Is this because the results are truncated? It might be useful to show the results more completely.

(2) It seems odd that the initial motor bias for small rPT was absent for pro-saccade trials (Figure 5d). Is there any explanation?

(3) The Introduction is devoted almost entirely to describing one or two past studies by the current authors. There is substantial literature on anti-saccades in humans and NHP. The Introduction should orient the reader to this literature and set up the questions addressed in the current study. As it stands, the Introduction does little to frame the issues for a general audience, or even for readers who specialize in oculomotor behavior and neurophysiology.

*Reviewer #2 (Recommendations for the authors):*

Line 1. "Oculomotor circuits generate eye movements based on the physical salience of objects and current behavioral goals, exogenous and endogenous influences, respectively." This is an interesting hypothesis, but it should not be stated as a matter of fact. Salience and behavioral goals are only some of the factors that influence eye movements.

Line 18. "The oculomotor system of primates chooses a new target to look at every 200-250 ms." Is there literature to support this? Are all eye movements the result of a "choice?"

Line 20. "Physical salience" is a questionable construction. Salience can be computed from physical properties of the stimulus, but that doesn't mean it is itself a physical property.

Line 28. Please change “commands” to “instructs”.

Line 34. Plotting accuracy as a function of reaction time is not a new concept. It's a conventional speed-accuracy trade-off. Inventing new terms for established ideas is not helpful. Chronometry or chronometric function is a conventional term for relating reaction time to performance, or, in general, for discussing the time dependence of mental processes. Unless a compelling case can be made for distinguishing "tachometry" from "chronometry," please use the latter.

Line 37. "Attentional vortex" seems like a poor word choice. Is there a rotational component to this phenomenon?

Line 46. It is remarkable that the term "race-to-threshold" could be used without a single reference to the vast literature dealing with this class of model. For example, Noorani and Carpenter 2011 adapted the LATER model to predict error rates v. reaction time for anti-saccades. Their model had 3 ramp-to-threshold processes, so it was basically a race model. Reviewed here: https://doi.org/10.3389/fnint.2014.00067

Line 55. This compound sentence is very difficult to parse.

Line 68. How are "compelled" movements related to "urgent" movements. Also, please change "compel" to "instruct" or something more appropriate. Human subjects can't be compelled.

Figure 2 a/e. It appears that the saccades in these examples would be initiated before the cue appears, i.e. they have a negative "raw processing time." Were such saccades actually observed in the human data? If not, what held subjects in check during the gap interval.

Figure 2d. If the height of the gray rectangles is different, can one assume that the proportion of pro:anti saccades was not 50:50? This isn't stated.

Behavioral task:

What was the criterion for classifying a trial as correct or incorrect? Was every trial scored as correct or incorrect, or were there other outcome categories, e.g. "no response" if the time limit was exceeded? If other outcomes were considered, how were these incorporated into the calculation of percent correct?

It would be nice to show the effect of gap duration on performance, both in terms of fraction correct and response time distributions.

Model:

It isn't clear how the cue is modeled or how it interacts with the endogenous preparatory signals. Is the cue represented by a square wave pulse? Some other function? Does it interact additively or multiplicatively? How exactly does it cause the endogenous signals to accelerate or decelerate?

Does the model account for reaction time distributions?

Line 408. Are the analytical functions in equations 1-4 derived from the race model? Why are fits shown in Figure 5, but not Figures3 and 4?

Are there any figures showing the race model fit to the empirical data?

Discussion:

Line 263. "We used the contrast between urgent pro- and antisaccade performance to test the idea that exogenous and endogenous contributions to saccadic target selection are largely independent." Given equal prior probabilities of A and B in an AB binary choice, is there any plausible scenario in which the endogenous and exogenous contributions would be inter-dependent? Can the authors provide an example?

Line 314. "exogenous response?" Did the authors mean "response to the exogenous cue?" It isn't clear that a response can ever be exogenous.

Line 308. The section that starts here undermines the conceptual clarity of the model. As this reviewer understands it, inhibition in the model serves to decelerate endogenous movement plans, as stated on line 323. This section packs in a lot of ideas that seem intended to clarify the idea of response inhibition, but it needs more work.

The "exogenous capture" described by the authors seems reminiscent of that reported by Bisley and Goldberg 2003.

How do the current results relate to the literature on counter-manding, which employs very similar race models.

*Reviewer #3 (Recommendations for the authors):*

Lines 82-83: Is the "exogenous response interval" (ERI) in this manuscript the equivalent of the "attentional vortex" in Stanford et al., 2019 *eLife*?

Lines 319 and 322: Everling's last name is misspelt ("Evering") for his 2000 paper with Munoz.

Line 353: Were any of the subjects authors?

Line 400: I might have missed this in previous papers…is there evidence that the behavioral effects scale with gap duration for urgent gaps? I.e., does a gap of -200 ms produce larger effects than a gap of -100 and/or 0 ms?

---

## [Author Response]

Essential revisions:(1) For the trials with low luminance, the asymptotic level of accuracy remained quite low (Figure 4e). Is this because the results are truncated? It might be useful to show the results more completely.

We truncated the results to highlight the precise bifurcation points of the high versus low luminance curves; however, indeed, truncating the curves cuts off the late rise in antisaccade performance. The full curves for these low luminance trials are shown above. The low luminance curves reach the same level of asymptotic accuracy as the high luminance curves, albeit ~40 ms later. The figure caption now notes the reason for using a limited rPT range in this case, and the full curves for the 3 participants in this experiment are now shown in a supplementary figure (Figure S3).

(2) It seems odd that the initial motor bias for small rPT was absent for pro-saccade trials (Figure 5d). Is there any explanation?

We do not know for sure, but there are two things to note about the motor biases. One is that they are quite idiosyncratic (see Figure S4); participants may demonstrate a strong bias in both tasks, in one of them, or in neither. In Figure 5, data are pooled across participants, so the individual extremes are hidden, but a couple of subjects did show strong biases in prosaccade trials in the condition of Figure 5d. So, the effect was definitely present in some cases. The second general observation is that prosaccades are perceptually easier than antisaccades. Participants can go along with the exogenous pull of the cue and, in effect, have more time to make a prosaccade than an antisaccade. Perhaps this is partly why motor biases were generally weaker and less consistent in the prosaccade task compared to the antisaccade. All this, however, is quite speculative.

(3) The Introduction is devoted almost entirely to describing one or two past studies by the current authors. There is substantial literature on anti-saccades in humans and NHP. The Introduction should orient the reader to this literature and set up the questions addressed in the current study. As it stands, the Introduction does little to frame the issues for a general audience, or even for readers who specialize in oculomotor behavior and neurophysiology.

This being a Research Advance, the Introduction had been crafted with just the essential context (which was provided by the earlier article), but the point is well taken; it becomes an easier read with a broader scope if this literature is acknowledged. The new second paragraph of the Introduction now highlights some of this important prior work, and puts ours in a wider context.

Reviewer #2 (Recommendations for the authors):Line 1. "Oculomotor circuits generate eye movements based on the physical salience of objects and current behavioral goals, exogenous and endogenous influences, respectively." This is an interesting hypothesis, but it should not be stated as a matter of fact. Salience and behavioral goals are only some of the factors that influence eye movements.

The sentence was edited to make it clear that exogenous and endogenous influences are not the only factors driving eye movements.

Line 18. "The oculomotor system of primates chooses a new target to look at every 200-250 ms." Is there literature to support this? Are all eye movements the result of a "choice?"

We’re not quite sure what is the objection to the term “choice” here, as it simply refers to the selection of one of many alternatives. We now say that “the oculomotor system specifies a new target to look at,” which is perhaps more neutral.

Line 20. "Physical salience" is a questionable construction. Salience can be computed from physical properties of the stimulus, but that doesn't mean it is itself a physical property.

Thanks for noting this. We now refer simply to “salience,” pointing out that it typically depends on key physical properties of the stimulus.

Line 28. Please change “commands” to “instructs”.

This has been changed.

Line 34. Plotting accuracy as a function of reaction time is not a new concept. It's a conventional speed-accuracy trade-off. Inventing new terms for established ideas is not helpful. Chronometry or chronometric function is a conventional term for relating reaction time to performance, or, in general, for discussing the time dependence of mental processes. Unless a compelling case can be made for distinguishing "tachometry" from "chronometry," please use the latter.

While the chronometric function is an established convention for relating RT to performance, it does not fit our analyses here. In general, RT and processing time are different, and more to the point, the curve relating performance to processing time has vastly different properties from that based on RT; in particular, it eliminates the speed-accuracy tradeoff to yield a behavioral metric that fundamentally reflects perceptual capacity only (Salinas et al., 2014, 2019; Salinas and Stanford, 2021; Stanford and Salinas, 2021). For that reason, a new naming convention needed to be established, and was done some time ago (Stanford et al., 2010).

Line 37. "Attentional vortex" seems like a poor word choice. Is there a rotational component to this phenomenon?

The mention to that term was eliminated.

Line 46. It is remarkable that the term "race-to-threshold" could be used without a single reference to the vast literature dealing with this class of model. For example, Noorani and Carpenter 2011 adapted the LATER model to predict error rates v. reaction time for anti-saccades. Their model had 3 ramp-to-threshold processes, so it was basically a race model. Reviewed here: https://doi.org/10.3389/fnint.2014.00067

The race is a useful analogy for the oculomotor mechanism that triggers saccades, but this is simply an experimental fact that applies to any saccadic choice. Prior antisaccade models, including the LATER implementation, are of little relevance to the work presented here because they do not consider the urgent regime. This situation is very different from the standard, non-urgent experiment. In any case, we now use the term “saccade competition model,” to avoid unintended implications.

Line 55. This compound sentence is very difficult to parse.

We tweaked the sentence, which we think is more clear now.

Line 68. How are "compelled" movements related to "urgent" movements. Also, please change "compel" to "instruct" or something more appropriate. Human subjects can't be compelled.

The “compelled” task naming convention has been previously published, dating back to Stanford et al., 2010, and in numerous subsequent publications. To compel someone is to pressure them into doing something they wouldn’t necessarily do spontaneously. We believe “compelled” is a highly appropriate term to use here because it conveys the somewhat unintuitive nature of the task, namely, that one must respond before knowing what the correct answer is, something that wouldn’t normally happen unless the right task conditions are in place.

Figure 2 a/e. It appears that the saccades in these examples would be initiated before the cue appears, i.e. they have a negative "raw processing time." Were such saccades actually observed in the human data? If not, what held subjects in check during the gap interval.

Indeed, saccades with negative raw processing times were observed in the human data. For long gap lengths, participants sometimes respond before the cue is revealed (as intended). Figures report only saccades with positive processing times because, while they do occur, such early guesses are infrequent and not particularly informative.

Figure 2d. If the height of the gray rectangles is different, can one assume that the proportion of pro:anti saccades was not 50:50? This isn't stated.

The proportion of pro:anti saccades was 50:50, as is now stated in the Methods. Note, however, that even if the proportion had been different, this would have had no effect on the chance level. The figure caption now specifies that chance was 50% in both tasks.

Behavioral task:What was the criterion for classifying a trial as correct or incorrect? Was every trial scored as correct or incorrect, or were there other outcome categories, e.g. "no response" if the time limit was exceeded? If other outcomes were considered, how were these incorporated into the calculation of percent correct?

Correct versus incorrect saccades were scored based on the direction of the first saccade made after the go signal. Completed trials were scored even if they exceeded the time limit; to the participants, going past the time limit simply resulted in different feedback (no sound). Aborted trials, including those with fixation breaks or primarily vertical saccades, were excluded from analysis. These details are now included in the Methods.

It would be nice to show the effect of gap duration on performance, both in terms of fraction correct and response time distributions.

The effect of the gap has been documented at length in previous publications, and the bottom line is that, once processing time has been taken into account, the effect of the gap on performance is very small. This is now mentioned around line 188. We think elaborating too much on the gap would be rather distracting of the key findings, which focus on processing time. However, we appreciate that similar questions may occur to other readers (e.g., see the last comment from Reviewer #3), so we have added a supplementary figure that illustrates the strong invariance of the tachometric curve relative to gap sampling (Figure S3).

Model:It isn't clear how the cue is modeled or how it interacts with the endogenous preparatory signals. Is the cue represented by a square wave pulse? Some other function? Does it interact additively or multiplicatively? How exactly does it cause the endogenous signals to accelerate or decelerate?

The model is described in detail in Salinas et al., (2019), and is very similar to prior instantiations designed to replicate urgent choice tasks (Salinas et al., 2010; Stanford et al., 2010; Shankar et al., 2011). For the purposes of this model, there is no function representing the cue itself. Rather, we consider the effect that the cue must eventually exert on the motor plans. We assume that the cue information arrives at the motor planning circuit (akin to, say, FEF) at a certain moment in time, at which point the plans start being modulated accordingly; so, thought of as a function of time, this would correspond to a step function representing the time at which the acceleration and deceleration turn on.

Does the model account for reaction time distributions?

Yes, it reproduces with remarkable detail the RT distributions for correct and incorrect trials, which have a very rich structure. This has been documented for similar urgent tasks (Salinas et al., 2010; Stanford et al., 2010; Shankar et al., 2011; Salinas and Stanford, 2013) and for the compelled antisaccade task in particular (Salinas et al., 2019). This is now mentioned around line 110.

Line 408. Are the analytical functions in equations 1-4 derived from the race model?

No, they are entirely independent. They are simply a means to quantify perceptual performance, i.e., to determine key characteristic metrics from the empirical tachometric curves, which are noisy. This is now stated in the Methods, just before the equations. Note, however, that the fits represent only some of the methods used for analysis of the data.

Why are fits shown in Figure 5, but not Figures3 and 4?

Figure 3 shows simulated tachometric curves. No fits are necessary to determine their characteristic metrics. In Figure 4, none of the key metrics derived from the fits are shown or discussed; that just wasn’t necessary. Analyses based on the fits produced consistent results, though.

Are there any figures showing the race model fit to the empirical data?

In Salinas et al., (2019), we did precisely this type of analysis on an individual participant basis for the CAS model. In the current study, the model fits are also good but they do not add anything critical. The key here is the qualitative difference between the pro- and antisaccade conditions, which is essentially independent of any parameter settings. So, in short, yes there are, but we think they would be more of a distraction than anything else.

Discussion:Line 263. "We used the contrast between urgent pro- and antisaccade performance to test the idea that exogenous and endogenous contributions to saccadic target selection are largely independent." Given equal prior probabilities of A and B in an AB binary choice, is there any plausible scenario in which the endogenous and exogenous contributions would be inter-dependent? Can the authors provide an example?

Interactions between exogenous and endogenous contributions might have been expected based on several arguments, and could have manifested in many ways. The exogenous response in anti trials could have been strongly attenuated, strongly delayed, or much less sensitive to luminance relative to that for the pro, but the differences were actually minimal. The exogenous capture during anti trials could have been much stronger in interleaved as compared to blocked conditions (due to adaptation or switching effects). Similarly, the capture could have been much stronger when the ongoing motor activity was in the direction of the cue location rather than in the opposite direction (due to attention enhancing visual sensitivity). Any one of these effects would have meant that the exogenous response was, in some way, modulated by non-sensory factors, but in fact, the results were consistent with it being largely impervious to all such conditions. We are looking at a signal that is highly localized in time (rPT ~90-140 ms) and that hadn’t been examined in isolation before (separately from antisaccade errors at long rPTs), so this was by no means a foregone conclusion. The second paragraph of the Discussion now makes this point.

Line 314. "exogenous response?" Did the authors mean "response to the exogenous cue?" It isn't clear that a response can ever be exogenous.

A response (whether neuronal or behavioral) can be driven exogenously or endogenously. “Exogenously driven response” has been purposefully shortened to “exogenous response,” as is now noted in the last paragraph of the Introduction, line 78.

Line 308. The section that starts here undermines the conceptual clarity of the model. As this reviewer understands it, inhibition in the model serves to decelerate endogenous movement plans, as stated on line 323. This section packs in a lot of ideas that seem intended to clarify the idea of response inhibition, but it needs more work.

In a sense, the objective of the section is just the opposite! — to point out that the idea of response inhibition is somewhat ill-defined. More specifically, it is not at all clear what, if anything, is being inhibited during anti trials that isn’t during pro trials. This section was heavily revised, so hopefully the message is more evident now.

The "exogenous capture" described by the authors seems reminiscent of that reported by Bisley and Goldberg 2003.

Yes, we believe they are related. Bisley and Goldberg analyzed activity from visually responsive neurons in area LIP that are strong candidates for mediating the sorts of exogenously driven behavioral effects discussed here. We now refer to this work in the Discussion, line 320.

How do the current results relate to the literature on counter-manding, which employs very similar race models.

This issue is discussed at length in a previous analysis of the countermanding task (Salinas and Stanford, 2013), which shows that much of the phenomenology in that task can be understood in terms of processing time. There are likely important mechanistic similarities between the countermanding and antisaccade tasks (for example, compare our data to Bompas et al., 2020), however, the exogenous contribution in the former is harder to isolate because so much of the dynamics remains covert. For that reason, the parallels are somewhat speculative. However, we have included our countermanding paper in the references mentioned when discussing similar mechanisms, line 362.

Reviewer #3 (Recommendations for the authors):Lines 82-83: Is the "exogenous response interval" (ERI) in this manuscript the equivalent of the "attentional vortex" in Stanford et al., 2019 eLife?

The ERI and the attentional vortex are not the same, but they are closely related. Both here and in Salinas et al., (2019), the ERI is defined as the time during which the motor competition is biased in favor of the cue location — in the model — so it is essentially a model parameter. In contrast, the vortex was meant to delineate the range of rPTs where captures are highly likely based on the tachometric curve, so it is an empirical construct. However, there is a direct connection: the longer the ERI, the deeper and wider the vortex. Here, we have removed the term “attentional vortex,” as it wasn’t critical and may have rubbed some readers the wrong way.

Lines 319 and 322: Everling's last name is misspelt ("Evering") for his 2000 paper with Munoz.

Thanks for noting the typo. It has been fixed.

Line 353: Were any of the subjects authors?

No, none of the subjects were authors.

Line 400: I might have missed this in previous papers…is there evidence that the behavioral effects scale with gap duration for urgent gaps? I.e., does a gap of -200 ms produce larger effects than a gap of -100 and/or 0 ms?

Essentially no, and that is one cool property of the tachometric curve! In the urgent range, the dependence of accuracy on time (rPT) is largely the same regardless of the gap duration (only a tiny deviation is generally found as one approaches gap=0). Each gap samples one relatively limited part of the same tachometric curve, and when all the gap snapshots are combined, this curve is revealed to its full extent. We have examined the dependence on gap in different ways in previous papers, but here we have added a supplementary figure that most directly illustrates the answer to the question posed by the reviewer (Figure S3).

References

Bompas A, Campbell AE, Sumner P (2020) Cognitive control and automatic interference in mind and brain: A unified model of saccadic inhibition and countermanding. Psychol Rev 127:524-561.

Salinas E, Scerra VE, Hauser CK, Costello MG, Stanford TR (2014) Decoupling speed and accuracy in an urgent decision-making task reveals multiple contributions to their trade-off. Front Neurosci 8:85.

Shankar S, Massoglia DP, Zhu D, Costello MG, Stanford TR, Salinas E (2011) Tracking the temporal evolution of a perceptual judgment using a compelled-response task. J Neurosci 31:8406-8421.

Salinas E, Stanford TR (2013) The countermanding task revisited: fast stimulus detection is a key determinant of psychophysical performance. J Neurosci 33:5668.

Salinas E, Stanford TR (2021) Under time pressure, the exogenous modulation of saccade plans is ubiquitous, intricate, and lawful. Curr Opin Neurobiol 70:154-162.

Salinas E, Shankar S, Costello MG, Zhu D, Stanford TR (2010) Waiting is the hardest part: comparison of two computational strategies for performing a compelled-response rask. Front Comput Neurosci 4:153.

Salinas E, Steinberg BR, Sussman LA, Fry SM, Hauser CK, Anderson DD, Stanford TR (2019) Voluntary and involuntary contributions to perceptually guided saccadic choices resolved with millisecond precision. *eLife* 8.

Stanford TR, Salinas E (2021) Urgent decision making: resolving visuomotor interactions at high temporal resolution. Annu Rev Vis Sci 7:323-348.

Stanford TR, Shankar S, Massoglia DP, Costello MG, Salinas E (2010) Perceptual decision making in less than 30 milliseconds. Nature Neuroscience 13:379-385.